# The Critical Role of Penicillin in Syphilis Treatment and Emerging Resistance Challenges

**DOI:** 10.3390/diseases13020041

**Published:** 2025-01-31

**Authors:** Arun Kumar Jaiswal, Lucas Gabriel Rodrigues Gomes, Aline Ferreira Maciel de Oliveira, Siomar de Castro Soares, Vasco Azevedo

**Affiliations:** 1Laboratory of Cellular and Molecular Genetics, Institute of Biological Sciences, Federal University of Minas Gerais, Belo Horizonte 31270-901, MG, Brazil; lucasgabriel388@gmail.com (L.G.R.G.); aline.fmo580@gmail.com (A.F.M.d.O.); 2Laboratory of Bioinformatics, Department of Microbiology, Immunology and Parasitology, Institute of Biological and Natural Sciences, Federal University of the Triângulo Mineiro, Uberaba 38025-180, MG, Brazil; siomars@gmail.com

**Keywords:** syphilis, *Treponema pallidum*, penicillin, antimicrobial resistance

## Abstract

Syphilis, a global healthcare burden, is a sexually transmitted infection caused by the spirochete *Treponema pallidum*, a spiral-shaped, Gram-negative obligate human pathogen. Despite its easy identification and treatability, the disease affects over 50 million people worldwide, with 8 million new cases in the 15–49 age group annually, as per the WHO 2024 report. If left untreated, syphilis progresses through its primary, secondary, latent, and tertiary stages, causing severe complications like neurosyphilis, congenital syphilis, and organ damage. The first-line treatment, penicillin, faces challenges, including logistical issues, shortages, allergic reactions, and patient non-compliance. Secondary treatment options are sparse, and there are reported cases of *T. pallidum* strains resistant to those antibiotics. The absence of an effective vaccine for syphilis has led to efforts to control its spread through sexual education, condom usage, and post-exposure prophylaxis with doxycycline, which raises concerns about antimicrobial resistance (AMR). The continued reliance on penicillin and the increasing rates of doxycycline post-exposure prophylaxis (DoxyPEP) use have both contributed to concerns about AMR development. Recent works pointing to emerging antibiotic resistance and treatment failures highlight the urgent need for new antibiotics to manage syphilis effectively and reduce dependency on penicillin. This review has focused on the shortcomings and limitations of penicillin treatment, recently identified antimicrobial-resistant strains of *T. pallidum*, and case studies where its application failed to treat the disease adequately.

## 1. Introduction

Syphilis is a sexually transmitted infection (STI) caused by the bacterium *Treponema pallidum*. It infects more than 50 million people worldwide [1] and 8 million people aged between 15 and 49 years according to the World Health Organization (WHO) 2024 report [2]. Penicillin is a recommended antibiotic as a first-line medication, and it has been formulated as long-acting benzathine penicillin G (BPG) to treat syphilis complications so far [3]. The disease exists in several stages, starting from primary, and then entering a second stage called secondary syphilis. If not treated well, the bacterium lives dormant in the host and prepares itself to reach the final stage of infection as tertiary syphilis [4]. The real challenges start if the patient is not treated well in the early phases of the infection (primary, secondary, and early latent), especially in the primary and secondary stages, which is pivotal to preventing long-term complications. The primary stage manifests as a painless sore (chancre) at the infection site (on the genitals and mouth), which can easily go unnoticed. Without proper medication, the infection enters the secondary stage with symptoms like rash, fever, and swollen lymph nodes [4,5]. Further, still-untreated disease in the secondary stage leads to the latent (early and late) stage, where the bacterium stays in the host body in a nonfunctioning dormant phase and does not show any further complications, with no clinical symptoms, in preparation for the tertiary stage. In tertiary syphilis, the bacterium *T. pallidum* is responsible for several chronic and life-threatening complications, such as neurosyphilis (multiple neurological complications—syphilitic meningitis, syphilitic myelitis, cerebral syphilitic gumma, atypical behavior, and neuropsychiatric symptoms), cardiovascular syphilis, oto/auricular syphilis, ocular and congenital syphilis, and even death [5]. Congenital syphilis is an outcome of untreated pregnant women who carry the bacterium *T. pallidum* and transfer it vertically through the placenta to their fetus (mother-to-child-transmission—MTCT) during pregnancy. This causes multiple clinical manifestations, including stillbirth and neonatal death, skin and visceral manifestations, and other asymptomatic infections in neonates [4]. Syphilis is curable if it is treated properly after the initial exposure of chancre on the genitals or mouth by giving antibiotic BPG intramuscularly [6], which is also the case for secondary and early latent phase (2.4 million units if no treatment has been started) and aqueous penicillin G, as 18–24 million units every day intravenously for 10 to 14 days is the recommended treatment for neurosyphilis according to the WHO guidelines for the treatment of syphilis [7]. Despite being treatable with penicillin, syphilis continues to be a global health concern with a significantly increasing number of cases. Several cases go unnoticed because of a lack of awareness, asymptomatic representations (especially in latency and tertiary), poor screening, the unavailability of screening materials and well-trained professionals (in low–middle-income countries and sometimes in developed countries as well) [4], and the taboo of talking about sexual health and behaviors in society [8]. Syphilis demonstrates a unique challenge among STIs due to its potential to mimic other diseases and its progression through the stages [9]. The increasing number of cases creates a global burden and requires proactive measures to enhance disease awareness, especially regarding the stages, diagnosis, and treatment [10]. Despite BPG’s effectiveness as a first-line treatment, the resurgence of syphilis cases globally points to gaps in public health responses and patient care [11]. One of the most serious issues is disease diagnosis during the latent phase (principally in the early latent phase), because of the asymptomatic behavior, and yet the bacterium remains active [5]. The early latent phase requires great attention in order for diagnosis, because once it is missed it often enables the infection to proceed to the tertiary stage, where patients can experience life-threatening manifestations, as mentioned above. Also, congenital syphilis and its rising cases promote the public health burden globally [12]. To deal with congenital syphilis, strengthening antenatal care and including syphilis screening in regular check-ups are essential to addressing this issue [13]. Improved diagnostic methods are required given the increasing number of cases [14]. Rapid tests have demonstrated promise in expanding access to syphilis testing, especially in low- and middle-income countries where laboratory infrastructure and resources are limited [14]. New rapid testing methods can provide more results at point-of-care units, help in making decisions for immediate treatment, and lower the disease-progression risks. However, the management of these tests continuously challenges that need global coordination. Furthermore, BPG shortages threaten global syphilis control efforts [15]. Fighting against the syphilis disease requires comprehensive approaches that should combine clinical care, public health interventions, and community engagement. Regarding the gaps in disease screening, ensuring the availability of effective treatments is critical to controlling the spread and minimizing the disease.

## 2. Treatment Regimen for Syphilis

Penicillin has been recommended as a first-line antibiotic for syphilis treatment (Figure 1) for many years and continues to be an effective treatment for *T. pallidum* infection. However, it still faces several restrictions and challenges while treating people, such as a lack of availability of penicillin at the primary health unit, allergies in patients [1], and sometimes the purity concentration of penicillin provided to the primary health unit [6,16]. Another challenge is the requirement, in certain cases, of intramuscular injection, which limits the availability of treatment to some patients [9]. According to the WHO, approximately 10–20% of people with syphilis refuse intramuscular injections, and healthcare providers are often adverse to applying them, due to the increased operational and personnel costs involved with them [7]. In recent years, we have found that huge shortages of benzathine penicillin G were reported to plague several low-to-middle income and developed countries between 2014 and 2016. These shortages, which are especially dangerous for congenital syphilis cases, have mainly impacted countries belonging to the African, South American, European, and Asiatic regions [3,17]. According to Nurse-Findlay et al., 39 countries out of a survey of 114 nations and territories reported shortages of BPG between those years [17]. In the South American continent, Brazil experienced significant BPG shortages during 2013–2017, resulting from market exits and quality disruptions, especially in the Rio de Janeiro municipality [18].

Second-line antibiotics such as doxycycline, ceftriaxone, and, in some special circumstances, azithromycin and erythromycin have been recommended to treat early syphilitic patients who are allergic and sensitive towards penicillin. Unfortunately, there are still restrictions on using these antibiotics in patient treatment [7]. In the case of doxycycline, the recommended dose is 100 mg taken orally twice daily for 14 days in early syphilis (defined as the primary, secondary, and early latent stages of infection) patients and 100 mg taken orally twice daily for 30 days in late (defined as more than 2 years without evidence of treponemal infection) syphilis or unknown syphilis [7]. Doxycycline is not suggested and recommended in pregnant women suffering from late syphilis and for congenital syphilis cases, because it can lead to severe adverse complications for the fetus or newborn as well as the lack of concrete experimental evidence for its effects on congenital syphilis infection [19]. Ceftriaxone is only recommended for early syphilis, with a dosage of 1 g taken intramuscularly once daily for 10–14 days, and is not recommended for late and congenital syphilis. In special circumstances, in cases where penicillin is not available or out of stock, a single 2 g dose of azithromycin, taken orally, is suggested for early syphilis, but is not recommended for late and congenital syphilis [7]. Erythromycin is not recommended for early syphilis and congenital syphilis, except in the cases of late and unknown syphilis, with a recommended dosage of 500 mg taken orally four times daily for 30 days. However, the WHO recommends caution in the case of erythromycin usage due to the low quality of evidence regarding its effectiveness [7]. The usage of amoxicillin is not included in the WHO guidelines for syphilis treatment; however, recent evidence has pointed to its effectiveness at treating the infection. Amoxicillin was shown to be effective as a monotherapy with a dosage of 1.5 g/day, taken orally for four weeks, and in conjunction with probenecid, with a dosage of 3.0 g/day, taken orally for two weeks [20].

In the recent past, doxycycline has been evaluated as a prophylactic medication against STIs and syphilis as pre-exposure and post-exposure prophylaxis in high-risk populations, such as men who have sex with men (MSM), female sex workers (FSW), and patients with HIV (PrEP, before the exposure against *T. pallidum*, or PEP, after potential exposure of syphilis within 24–72 h after sexual activity, respectively) [21,22]. Some recently published guidelines and editorials provided by scientists and experts have raised comments and concerns on how the widespread and long-term use of doxycycline as a prophylactic medication for STIs can lead to resistance against STI-causing pathogens [23]. In addition, the regular use of antibiotics as prophylactic medication may disrupt the usual microbiome of the body and can cause negative health impacts [24]. There is strong evidence of the effectiveness of doxycycline PEP (DoxyPEP), particularly in populations of MSM, in preventing syphilis infection. Thus, certain medical bodies such as the Australasian Society for HIV, Viral Hepatitis and Sexual Health Medicine (ASHM) have evaluated its benefits as outweighing the risks of developing AMR [21]. However, while no studies have pointed to doxycycline resistance in *T. pallidum* due to DoxyPEP so far, macrolide resistance has been documented to occur, indicating the potential for developing doxycycline resistance as well [21]

Macrolides (associated with changes at the target site through mutation or methylation) such as azithromycin and erythromycin and tetracyclines such as doxycycline are bacteriostatic antibiotics that restrict bacterial growth by interfering with bacterial protein production. Macrolides inhibit protein synthesis by binding to 23S rRNA of the 50S ribosomal subunit, associated with tetracyclines, and inhibit protein synthesis by binding to 16S rRNA of the 30S ribosomal subunit, respectively [25].

The WHO guidelines for syphilis treatment point to the need for developing new treatments capable of combating *T. pallidum* infection broadly in the cases of early, late, and congenital syphilis. Ideally, such a treatment should be a short-course, orally applied antibiotic capable of crossing both the blood–brain and placental barriers in order to effectively combat neurosyphilis and congenital syphilis [7].

## 3. Antimicrobial Resistance in *T. pallidum*

Recent research on the antibiotic susceptibility patterns of *T. pallidum* has pointed out that the bacterium remains susceptible to the most recommended syphilis treatments [26]. However, with recent reports of treatment failures, growing concerns about rising AMR, and the detection of variants resistant to antibiotics such as azithromycin, there is still cause for concern about developing resistance in this infection.

The first erythromycin treatment failures were observed in 1964 and 1976, the latter marking the first case where pregnant women gave birth to children with congenital syphilis. However, the association of this failure with erythromycin-resistant *T. pallidum* was not entirely congruent, since the ability of erythromycin to cross the placental barrier was not yet known [25,27]. In 1977, *T. pallidum* strain SS14 showed a high resistance level to erythromycin and cross-resistance to azithromycin [25]. Until then, the basis for this strain’s resistance to the macrolide class was not established. With the advent of genomic analyses, the hypothesis of resistance developed through the endogenous survival mechanisms (such as adaptation and cell death) of *T. pallidum*, leading to a mutation in the 23S rRNA gene, was proposed [25].

Later, azithromycin’s clinical failure was observed in San Francisco, California United States between 2000 and 2004. During this time, the prevalence of erythromycin-resistant *T. pallidum* infections increased significantly, suggesting that the mutant strain became more frequent [27]. This study confirmed that the antibiotic developed resistance to the A2058G mutation in the 23S ribosomal RNA (rRNA) gene of the bacterium, which was also observed in a high quantity of syphilitic patients on the west coast of the United States, Canada, and in Dublin [28,29,30]. The A2058G mutation alters the ribosome binding site where azithromycin binds, resulting in a conformational change in the ribosome, preventing the antibiotic from binding effectively and inhibiting protein synthesis [31].

In 2009, Matejková et al. discovered a new mutation, A2059G, in *T. pallidum* in syphilitic patients from the Czech Republic [31], and later on, the mutation was also reported in *T. pallidum* in syphilitic patients from England, the USA, and China [29,32,33,34]. In 2018, work published by Yuecui Li also found the mutations of A2058G and A2059G among the *T. pallidum* strains, and the mutation of A2059G showed higher resistance for azithromycin compared to A2058G mutations; this was possible because Yuecui Li and collaborators identified that when subtyping the TPR gene, which shows higher resistance to macrolides, the A2058G mutation was observed in 3 subtypes, while A2059G was present in 5 of the 8 determined subtypes by the group [34].

Recently, it was unfortunate to see the clinical failure case of penicillin against a 54-year-old African American male who had a neurosyphilis complication. The patient was treated with intravenous penicillin G, with 24 million units daily for 14 days. After two months, the patient was hospitalized for altered behavior and mental-status changes from the cognitive baseline, and the cerebrospinal fluid (CSF) rapid plasma reagin (RPR) titer was 1:4, the same as at the initial diagnosis. Although this report does not yet represent a case of resistance to benzathine penicillin G, we observe a worrying scenario, especially with studies reporting the existence of *T. pallidum* genetic variants linked to the binding protein of this drug, even though none of the identified variants so far have been able to significantly alter the bacterium’s response to penicillin [35,36,37].

Penicillin is a β-lactam antibiotic that disturbs the function of penicillin-binding proteins (PBPs) for bacterial lysis by inhibiting bacterial growth [38]. It is estimated that 10–20% of early syphilitic patients experience clinical failure. Considering all recent events, last year (in 2023, the first-ever clinical study was published demonstrating the treatment failure of penicillin in syphilitic patients and penicillin resistance-related gene mutations in early syphilitic cases), clinical failure was also observed in early syphilitic patients by identifying the non-synonymous single nucleotide polymorphism (SNP) gene mutation related to penicillin resistance against the bacterium *T. pallidum*. Mutation in the gene belongs to the family that encodes the penicillin regulatory proteins in Chinese strains of *T. pallidum* (TPANIC_0500, TPANIC_0760, and TPANIC_0705), which suggested that these SNPs may be linked to penicillin resistance in *T. pallidum* [39].

*T. pallidum* is a highly complex bacterium with numerous specific characteristics that pose a challenge for scientists. In addition to its ability to inactive beta-lactamase enzymes, which are essential for the efficacy of penicillin-based antibiotics and recently identified mutations, studies showed that *T. pallidum* can strengthen its bacterial capsule in response to inadequate drug exposure through the activity of surface proteins [40,41]. The outer membrane proteins on the surface facilitate the bacterium’s adhesion to host cells, secreting substances that neutralize the effect of antibiotics, thereby protecting the bacterium. Additionally, they can inhibit phagocytosis by evading the host’s immune system, interfering with the receptors of phagocytic cells, and preventing some of its antigens from being recognized by the immune system [42].

Back-to-back cases of clinical/treatment failure in neurosyphilitic patients and early syphilis reported in 2022 (USA) and SNPs in the Chinese *T. pallidum* strain in 2023 (China) suggest that in the future, the bacterium will be able to develop resistance against penicillin, and it will be interesting to see how and when. However, over the past 30 years, the global incidence of syphilis has increased by approximately 60%, from 8,845,220 cases in 1990 to 14,114,110 cases in 2019. This significant increase highlights the need to explore alternative antibiotic treatments, in addition to the recommended first-line and second-line antibiotics. After all, the excessive pressure on penicillin could induce the development of resistance. As we previously discussed, second-line antibiotics have several restrictions in the different stages of syphilis treatment, further emphasizing the importance of new therapeutic approaches to effectively and sustainably control the syphilis epidemic [43].

In a recently published randomized controlled trial, oral linezolid applied as a treatment for syphilis showed significant evidence in a rabbit model. The cure rate was 70% in the linezolid-treated group, while the BPG-treated group had a cure rate of 100%. However, the study was discontinued in non-pregnant adults with early syphilis, because the linezolid dose did not meet the non-inferiority criteria compared to BPG, indicating that a higher dose or an extended treatment regimen should be considered [1]. Table 1 summarizes the history of acquired resistance in *T. pallidum*.

## 4. Future Perspectives

Interestingly, the WHO has proposed strategies to reduce the number of cases of HIV, viral hepatitis, and STIs for the period of 2022–2030, where it intends to reduce the number of syphilis cases from 7.1 million to 5.7 by 2025 and to 0.71 million by 2030 [44]. Reaching the WHO goal will require significant effort and awareness of the disease in society, well-trained disease-screening professionals at the primary health units, the fast and accurate diagnosis of early phages of the infection, treatment, and recommendation of antibiotics with caution. Currently, an innovative method using cyclic voltammetry to detect the antigens and antibodies of *T. pallidum* has been developed by the Laboratory of Technical Innovation in Health (LAIS/UFRN) in partnership with Johns Hopkins University and Coimbra University. The prospects for implementing this method are positive, and it is expected to improve the diagnosis of syphilis, especially in hard-to-reach areas, contributing to the WHO’s goal of reducing the global incidence of syphilis and congenital syphilis cases to 50 or fewer per 100,000 live births in 80% of countries. It is worth noting that this project is still in the development and validation phase, meaning it has not been widely implemented in clinical practice [45]. Additionally, in 2024, a study conducted at Fiocruz Bahia demonstrated the excellent performance of the recombinant proteins TpN17 and TmpA in distinguishing between positive and negative samples of *T. pallidum*, suggesting a further increase in sensitivity through antigenic mixtures, which will be the next focus of the group’s investigation [46]. Even though there is no defined proposal, hope lies in the ability of science to find solutions to combat this infection.

Thus, it is important to emphasize that scientists should focus on new alternative antibiotics research by applying the latest strategies such as in silico drug discovery and virtual screening for compound molecule evaluation instead of performing the conventional antibiotic discovery strategies, as well as developing straightforward techniques for in vivo experiments to check the antibiotic susceptibility against *T. pallidum*. Indeed, these conventional strategies are helpful, as shown by the CeBra study, led by the WHO in collaboration with Brazilian universities, which aims to assess the efficacy of cefixime, an FDA-approved antibiotic for gonorrhea, as an alternative treatment for syphilis. Cefixime’s potential for oral administration makes it a promising option, especially in resource-limited settings where penicillin’s intramuscular administration poses challenges [47]. However, while the CeBra study is a valuable step toward expanding treatment options, it should be viewed as a preventive measure. The focus should shift toward developing more innovative treatments that fully address the limitations of current therapies, particularly for late-stage syphilis and cases involving neurological involvement, where drug efficacy is limited. Exploring novel antibiotics or even vaccine development is critical for long-term syphilis management.

Regarding the current evidence and scenario of increasing cases, clinical failure observation, the identification of resistance genes against bacterium, existing restrictions while treating with first-line, second-line, and prophylactic medication, and the halted research studies firmly underscores the need for new antibiotics against syphilis; instead of focusing on antibiotics that may treat multiple STIs, scientists should focus on antibiotics that may be disease-specific. As with treating syphilis, we should also focus on finding specific antibiotics that should help in early, neonatal syphilis, neurosyphilis, and syphilis in pregnant women, and may help to reduce the burden of penicillin treatment against syphilis.

## 5. Conclusions

The continued reliance on penicillin as the first-line treatment for syphilis is a factor of its historical significance and clinical success over the decades. It has been the gold standard in managing *T. pallidum* infections since its introduction in 1943 due to its high efficacy and relatively low cost [3]. However, as syphilis resurges globally, particularly in specific high-risk populations, the limitations of penicillin as a sole therapeutic agent are becoming more apparent [3].

Among the challenges presented by the continuous usage of penicillin are penicillin allergies, patient rejection due to a need for intramuscular injection, and supply-line troubles, limiting access to penicillin [1,3]. For those who cannot tolerate penicillin, alternative treatments such as azithromycin, doxycycline, or ceftriaxone are necessary. These alternatives, while effective in some cases, may not always offer the same reliability, mainly for treating neurosyphilis, or may be inapplicable during pregnancy, where alternatives like doxycycline cannot be used due to potential teratogenicity [1]. Addressing penicillin allergies through desensitization protocols has been proposed, but this process can be time-consuming and inaccessible in many settings, especially in resource-limited areas [1,3].

Though *T. pallidum* has remained susceptible to penicillin for decades, recent reports of potential resistance, although rare, raise concerns about the future of this treatment [29,30,31,32]. While potential penicillin-resistant syphilis strains are uncommon, the potential for reduced susceptibility or treatment failures is due to suboptimal drug levels, especially in latent syphilis or neurosyphilis. This factor points to the need for heightened vigilance, particularly as putative penicillin resistance mutations have been detected in the bacterium’s genome [38,39,40,41]. Reinfections and treatment failures are also on the rise, particularly in populations with high rates of HIV co-infection, further complicating treatment outcomes [36]. These issues suggest that we may be approaching a time where reliance on a single antibiotic regimen, even one as historically successful as penicillin, may not be sufficient.

From a public health perspective, the limitations of penicillin underscore broader challenges in addressing the syphilis epidemic [2]. Rising global rates, particularly in marginalized and high-risk populations, emphasize the need for a more comprehensive approach to syphilis management [43]; this includes better access to testing, treatment, and education in vulnerable communities. Populations with higher rates of reinfection, such as those with multiple sexual partners or substance-use disorders, pose a particular challenge to syphilis control efforts [42]. In these contexts, public health strategies must evolve to incorporate enhanced outreach, partner notification, and preventive interventions such as DoxyPEP, which is effective in reducing syphilis transmission in at-risk populations, although it raises concerns over potential resistance development [21,22,44].

In conclusion, while penicillin remains a powerful and essential tool in the fight against syphilis, its limitations cannot be ignored. Challenges such as allergic reactions, shortages, potential resistance, and treatment failures highlight the need for ongoing innovation in clinical practice and public health strategy. Future research must focus on identifying new therapeutic options, enhancing treatment adherence, developing new diagnostic and prophylactic tools, and developing public health programs that target at-risk populations more effectively. Only by addressing these multifaceted challenges can we ensure continued progress in controlling and ultimately eradicating syphilis in the modern era.

## Figures and Tables

**Figure 1 diseases-13-00041-f001:**
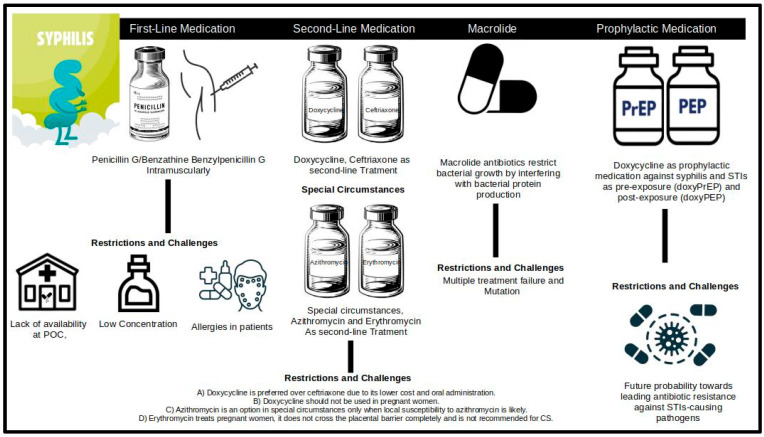
Antibiotic recommendations for syphilis in adults, adolescents, and congenital syphilis with restrictions and challenges. POC: point-of-care; CS: congenital syphilis [7].

**Table 1 diseases-13-00041-t001:** Acquired resistance of *T. pallidum* to recommended treatment drugs.

Acquired Resistance of *T. pallidum* to Recommended Treatment Drugs		
Treatment	Type of Resistance or Hypotheses	Year
MACROLIDES	First clinical failure.	1964 [25]
	First case of congenital syphilis.	1976 [27]
Erythromycin	Identified 23S rRNA gene mutation in the SS14 strain of *T. pallidum*.	1977 [25]
Azithromycin	A2058G mutation identified, a cross-mutation of the 23S rRNA gene from 1977	2000–2004 [28]
Spiramycin	A new 23S rRNA gene mutation, A2059G, was discovered.	2009 [32]
BETA-LACTAMS	First clinical failure was initially not linked to resistance.	2022 [36]
Benzathine Penicillin G (Example: Methicillin)	First penicillin failure.	2023 [39]
	Non-synonymous single nucleotide polymorphism (SNP) mutation, encoding penicillin regulatory proteins, found in Chinese strains.	
Penicillin-based drugs	Essential M proteins: strengthening of the bacterial capsule; increased adhesion capacity; immune system evasion	2021–2022 [40,41,42]

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
