# Peer review of "The Critical Role of Penicillin in Syphilis Treatment and Emerging Resistance Challenges"

_diseases, 2025, doi:10.3390/diseases13020041_

Round 1
Reviewer 1 Report
Comments and Suggestions for Authors
Even if SNIP is present in the gene for a penicillin-binding protein, I don't think that the function of the protein is completely lost, and I get the impression that there is an excessive amount of publicity about the fear of penicillin-resistant syphilis bacteria, even though there is no clear existence of penicillin-resistant syphilis bacteria.
Author Response
Comments and Suggestions for Authors
Even if SNIP is present in the gene for a penicillin-binding protein, I don't think that the function of the protein is completely lost, and I get the impression that there is an excessive amount of publicity about the fear of penicillin-resistant syphilis bacteria, even though there is no clear existence of penicillin-resistant syphilis bacteria.
Response: We thank the reviewer for the valuable feedback and perspective. Reviewer has raised an important point regarding the presence of SNPs (Single Nucleotide Polymorphisms) in penicillin-binding proteins and their potential impact on protein functionality. Indeed, it is crucial to acknowledge that the presence of SNPs alone may not necessarily result in a complete loss of protein function or resistance. Instead, they might alter penicillin's binding affinity or efficiency in certain cases, which justifies further investigation. We thank the reviewer for valuable feedback and perspective.
The intention behind emphasizing the possibility of penicillin resistance in Treponema pallidum was not to suggest that resistant strains are currently prevalent or well-established. Instead, it underscores the importance of vigilance in monitoring the evolutionary potential of Treponema pallidum under selective pressure, given the emergence of resistance in other bacterial species. Early detection of mutations associated with reduced susceptibility could guide proactive strategies to prevent resistance from becoming a clinical reality.
Furthermore, while penicillin remains highly effective against syphilis, the observed treatment failures in some cases (although rare) have raised concerns that merit careful evaluation. These could stem from improper dosing, reinfection, or potentially subtle genetic adaptations in the bacterium. Our goal is to promote continued awareness and research in this area to ensure the long-term efficacy of this critical antibiotic.
In addition, as we have highlighted in the review. BPG shortages have historically had grave impacts on vulnerable populations. In contrast, alternatives have their own concerns, such as the unavailability of pregnant patients and resistant variants, which points to a need for further research.
The reviewer’s observation reinforces the need for balanced communication that avoids overstating the threat while maintaining vigilance against potential resistance. We greatly appreciate your insights and the opportunity to clarify this aspect of the manuscript.
Reviewer 2 Report
Comments and Suggestions for Authors
Jaiswal et al. address an important topic regarding the critical role of penicillin in syphilis treatment and the emerging challenges of resistance. However, before considering the manuscript for publication, the authors should address the following issues:
- The article contains several grammatical errors, particularly with capitalization, such as in L12 ("Sexually Transmitted Infection"), L23 ("Doxycycline and Antimicrobial Resistance"), L52 ("Neurosyphilis"), and others.
- L20: "T. pallidum" should be italicized. Please check this throughout the manuscript.
- L24: What is DoxyPEP?
- L51 and throughout the manuscript: Please replace "Treponema pallidum" with "T. pallidum" as the full name was already provided earlier.
- Figure 1 legend: Please correct the capitalization (each word starts with a capital letter) and add a reference.
- Please add a title for Table 1 and provide references for each row.
- L251: What is LAIS/UFRN? Also, for all abbreviations, please provide the full name when they are first mentioned.
Author Response
Comments and Suggestions for Authors
Jaiswal et al. address an important topic regarding the critical role of penicillin in syphilis treatment and the emerging challenges of resistance. However, before considering the manuscript for publication, the authors should address the following issues:
Response: We thank the reviewer very much for the comments and suggestions. All the points mentioned by the reviewer have been updated in the revised manuscript.
-
The article contains several grammatical errors, particularly with capitalization, such as in L12 ("Sexually Transmitted Infection"), L23 ("Doxycycline and Antimicrobial Resistance"), L52 ("Neurosyphilis"), and others.
Response: We thank the reviewer for the comment. Correction have been done in the revised manuscript.
-
L20: "T. pallidum" should be italicized. Please check this throughout the manuscript.
Response: We thank the reviewer for the comment. Correction have been done in the revised manuscript.
-
L24: What is DoxyPEP?
Response: We thank the reviewer for the comment. An explanation of DoxyPEP has been added to the revised manuscript.
-
L51 and throughout the manuscript: Please replace "Treponema pallidum" with "T. pallidum" as the full name was already provided earlier.
Response: We thank the reviewer for the comment. Correction have been done in the revised manuscript.
-
Figure 1 legend: Please correct the capitalization (each word starts with a capital letter) and add a reference.
Response: We thank the reviewer for the comment. Correction have been done in the revised manuscript.
-
Please add a title for Table 1 and provide references for each row.
Response: We thank the reviewer for the comment. Correction have been done in the revised manuscript.
-
L251: What is LAIS/UFRN? Also, for all abbreviations, please provide the full name when they are first mentioned.
Response: We thank the reviewer for the comment. Correction have been done in the revised manuscript.
Reviewer 3 Report
Comments and Suggestions for Authors
This review is lenghty and verbose, despite that it misses mentioning therapy with 1.5 g/day oral amoxicillin and high dose oral amoxicillin plus probenecid. Moreover, it is not clear why it does not refer to the new researches about antimicrobial susceptibility of Treponema pallidum expanding the search of new antibiotics against syphliis.
Author Response
Comments and Suggestions for Authors
This review is lenghty and verbose, despite that it misses mentioning therapy with 1.5 g/day oral amoxicillin and high dose oral amoxicillin plus probenecid. Moreover, it is not clear why it does not refer to the new researches about antimicrobial susceptibility of Treponema pallidum expanding the search of new antibiotics against syphliis.
Response: We thank the reviewer for the comments. We had not included treatment with amoxicillin monotherapy or in conjunction with probenecid as it was absent from the WHO guidelines for syphilis treatment (as published in 2016: https://www.who.int/publications/i/item/9789241549714). We have added a section for this treatment alternative in section 2, citing the work of Ikeuchi and collaborators on determining the efficacy and tolerability of this treatment. As for the more recent research on antimicrobial susceptibility, we thank the reviewer for the perspective. We have added a section on recent research showing that T. pallidum is still susceptible to most recommended antibiotics. However, we sought to focus on new prospects for resistance development in this bacterium rather than its continued susceptibility.
Round 2
Reviewer 3 Report
Comments and Suggestions for Authors
Accept in the present form